# Musing with Petric Bodies, Hanging on to Dear Life

Julieanna Preston 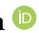

Toi Rauwharangi/College of Creative Arts, Te Kunenga ki Pūrehuroa/Massey University, Te Whanga-nui-a-Tara/Wellington 6041, Aotearoa/New Zealand; j.preston@massey.ac.nz

**Abstract:** *Musing with Petric Bodies, Hanging on to Dear Life* is an essay that critically reflects on the live performance work "Becoming Boulder", which occurred on 31 January 2015 as part of the Science Communication Art New Zealand Intercreate Symposium at New Plymouth, New Zealand. I performed a contact improvisation with a large andesite boulder, in a king tide, on a stormy day, at a culturally significant place for an extended period of time. Written using the present tense and as a dialogical text, the essay employs ekphrasis and practices geo-poetry to colour the scene and critically contextualise the potentials and limits of empathetic engagement with another form of organic assemblage. Complexities that come with being a foreigner or immigrant, well-versed in contemporary New Materialist discourse, and dwelling in a land rich with indigenous knowledge are voiced amongst gestures to get close to, identify with, and perform as an ancient, far from dead weight, body. While musing and critically contextualising on the potentials and limits of empathetic engagements, the essay seeks to exemplify the value of material situated learning that occurs in the space of making or doing of durational, experimental, site-responsive performance works.

**Keywords:** geo-poetry; geology; site-responsive performance; ekphrasis; geophilia

## 1. Being Here

I am writing this essay in the dappled shade of trees in Ōtaki Beach, a small township on the west coast of the North Island, Aotearoa in the watchfulness of Kāpiti Island. I can hear waves of the Tasman Sea crashing, smell the heavy perfume of noxious lupin blooms invading the seagrasses, and feel the rain clouds building against the Tararua Ranges to the east. A large male pheasant screeches an alarm while taking flight from the neighbourhood feral cats. Though technically a citizen, I am what is called a manuhiri, a visitor, to this land. Not born here, not indigenous, I am an immigrant, foreigner, and an alien, even after twenty-five years of living in this country, a place that is still very much grappling with what it means to be bi-cultural and how to decolonise or re-indigenise. In this place, one's history and relation to these histories are always front and centre.

I originate from the United States; my father was a Navy officer with ancestors from Wales and England. My mother was an English teacher and watercolour artist with genealogy stemming from Sweden and Croatia. We moved almost every two or three years, always to a seaport or near a water body, which fuelled my slight obsession to anything water-related. My mother left me with a love of writing and a strong sensibility towards colour. It is no wonder that these things factored into shifting house, family, and life to Aotearoa with its big seas and nuance shades of green. Being a visitor to a land that is not yours is not something that one overcomes easily, if at all. Rather, I am continuously coming to terms with this difference by shaping a positionality in relation to histories that I have previously taken for granted. For me, like many Pākehā—literally meaning not normal and translated as white European, or more precisely, "pale, imaginary beings resembling men" (Marcetic 2018)—I am a work in progress prompted by a commitment to Tangata Tiriti. This term refers to being an active citizen in learning about the Treaty of Waitangi and what it takes to be a good treaty partner. This involves being at peace with your position, respecting boundaries, being prepared to make sacrifices, do the work that is required of

yourself, standing up for Māori language, health, children, and women, and not expecting to be rewarded (Ngata n.d.). Creative works, such as "Becoming Boulder", are the way I learn to be in this place, its land, people, customs, and values. It is learning by doing, and in that sense, every live performance is an experiment that wonders about things through an embodied process more so than asserting knowledge that has already been acquired. Here, there is synergy with science and technology scholar Donna Haraway's term 'situated knowledge', which celebrates multiple feminist formations of knowledge that are contingent, as well as dependent, on partial and fragmentary situations to make a rich account of a close-at-hand and immediate local world (Haraway 1988). Haraway's concept of situated knowledges also applies to 'situated material learning', a mode of feminist practice that explores a multiplicity of on-going transformation of local conditions. "The lesson to be learnt here is: A practice is never independent of its environment-world or milieu, and you do not know in advance what a practice can become; it is a matter of experiencing-experimenting" (Frichot 2019, p. 133).

The quality of not knowing what will happen is complemented by an approach that embraces site-responsiveness over site-specificity. The former acknowledges the spatio-temporal contingency of a place; neither scripted or pre-determined, the performance unfolds as a relational encounter, in which discovery and gestures are integrated processes. In my practice, the site, its historic, political, and cultural context, and its materiality, are equal collaborators to how the encounter happens and what new understanding is produced. This runs contrary to site-specificity which, for me, objectifies place, demarcates it with hard boundaries and appears to instrumentalise its character and at the same time, regard it as a backdrop. In site-specific performance, the work happens in the place rather than with the space; "dominant positivistic formulations . . . are deemed to have reached a point of aesthetic and political exhaustion" (Kwon 1997, p. 1).

In "Becoming Boulder", I practiced approaching, meeting, greeting, and slowly getting to know a stranger; anything could happen. This encounter extends a system of movements called contact improvisation. Originated by choreographer Steve Paxton in the 1970s, contact improvisation is a dance form based on "the communication between two moving bodies that are in physical contact with one another and their combined relationship to the physical laws that govern their motion—gravity, momentum, inertia . . . Contact improvisations are spontaneous physical dialogues that range from stillness to highly energetic exchanges . . . a free play with balance . . . bringing forth a physical/emotional truth about a shared moment of movement that leaves the participants informed, centered, and enlivened" (Paxton 1979, p. 35). In the instance of this performance, haptic gestures afforded the transfer of energy as a form of non-verbal expressive communication between my live body and your body—the body of a boulder, which historically has been assumed to be simply dead matter. And yet, while both of our bodies are subject to entropy, I have a hunch that there is more to our chemistry than convention portends.

I used to believe that this performance started as I drove north towards New Plymouth where to participate in a three-week artist residency. I left before sunrise to stop at every beach along the two hundred and eighty kilometres. As I got closer to my destination, the sand turned black with iron and a snow-capped sharp peaked mountain commanded attention. I felt that I was entering deep time. Rocks, stones, and boulders live by this time; it is slow, the quintessential of duration. I now contend that the performance was seeded by the multiple encounters I have had with prominent volcanic mountains. When moving from Los Angeles to The Philippines in 1965, the cruise ship stopped in Yokohama, Japan. The sight of Mount Fuji was vast and spectacular despite the great distance between us. At seven years old, we visited Taal Volcano in Batabgas, Philippines. Standing at the rim, I was thinking how incongruous the story of its fiery destruction was to the thick blanket of moist clouds that were hiding its blown crown. The audible thresholds of thrashing, spewing, gently rocking sounded at odds with the heavy breath of dormant slumber. Later that summer, I swam in Tadlac Lake, a volcanic crater and much to my dismay, also colloquially known as Alligator Lake. Not being able to see below the surface

of the opaque grey-yellow-green water is still vivid. In 1992, I travelled to Klamath Falls in Oregon, USA to check on a piece of land I had been gifted. Ladened by thick ash and barren rock outcropping, small scraggly pines and wildflowers signalled the early stages of regeneration after the 1980 eruption of Mount Saint Helen's. In 2001, my partner trained in hot rock massage therapy. We collected the coveted dense, well-shaped basalt rocks from the northernmost Quaternary volcanic fields in Auckland. These rocks sit on my windowsill exuding calmness as I write. Their virtue to hold heat for long periods of time has also been the saving grace to cold feet between the winter sheets. These are but a few of the ways my life has been profoundly linked to rocks and their geo-matter.

*These are some of the things I brought with me to this encounter with you.*

## 2. Being There

The atmosphere and context of "Becoming Boulder" is intricately connected and dependent on the cultural, geological, material and social fabric of the place. Taranaki Maunga, Mount Taranaki, also known as Mount Egmont, is called a sleeping giant (NZ Herald 2021). Predicted to royally spew in the next fifty years, this 2500 m-high cone-shaped stratovolcano erupted in 1854, 1790, and 1655 "with widespread tephra falling across the central North Island" of Aotearoa (GeoNet n.d.). The debris of 130,000 years of intermittent volcanic activity litters the landscape at the base of this noble peak.

Huge round andesite boulders catapult from the belly of the mountain, into the sky, tempered and quenched. Their weighty mass lands in the creases and folds of the hills below, rolling into the rivers and streams as compact formations, cemented by sand, scoria, and ash. The sound is deafening. According to Aristotle, a foreigner to this land, rocks do not fall to the ground because of gravity; they fall because the earth is their natural place, it is their home (Robertson 2012, p. 98).

*Dear Boulder, this is your petrogenesis, your creation story, or at least one of them.*

Scientists estimate that you may be five billion years old in the making (National Geographic n.d.). In a pre-Newtonian cosmos, a rock flung upward "is a violent centrifugal urge, a natural tendency to flee earth's center and move to the terrestrial circumference as the place where it is happiest, most itself" (Allen 2012, p. 129). According to Te Ao Māori (world view), the god of magma Pūtoto is constantly seeking outward paths towards the Earth's surface. Along his trajectory, Pūtoto deposits many koha (gifts) for the guardians of the Earth's bedrock and crust. Through the natural processes of heating, compression, solidification, weathering, and erosion, new varieties of stones, rocks, sand, and minerals are generated out of these gifts (GNS Science n.d.).

Large chunks of basalt take flight amongst the molten rock, gases, and pumice.

*Your grey blue, pink, and pale-yellow tones help me distinguish you from your denser grey to black to brown geo-cousins.*

Worn by weathering, these expelled rocks blanket the mountain's western, southern, and northern faces of fertile land as aprons of iron rich black sand. Meanwhile, the Tasman Sea holds Aotearoa and Australia apart and together. It lashes the edges of both islands, eroding their low-lying river basins, claiming back land here, and redistributing it elsewhere. The hungry swells are unconcerned with buildings, infrastructure, or property-ownership. The sea's power grows stronger by the westerly onshore winds that beat the land into submission on a regular basis. Māori stories talk about Tangaroa, the sea god, nibbling the land with his moods of stormy swollen waters as an act of eternal conflict with his brother, Tūmatauenga, the god of warfare (Te Ara n.d.). Might versus mighty.

More than thirteen kilometres of seawall protect the shoreline between Bell Block and Port Taranaki in New Plymouth. The seawall's purpose to hold back the sea's wrath is rationalised as a relatively level serpentine coastal promenade for tourists and residents to walk, run, cycle, skate, and scoot with ease between beaches and cafes complemented by dramatic sea views—a western European urban design tactic promoting exercise, move-

ment, and leisure (New Plymouth District Council n.d.). Conservation turned capitalistic destination.

Constructed at the will of the city council and $25 million, this line of defence is broken in only two places, one of them at the heart of the urban fabric where the Huatoki Stream is liberated from its storm water concealment and let loose to join the ocean (Taranaki Daily News/STUFF 2022). In engineering terms, the rocks resist the thrust of eroding tides and manage the river mouth's desire to wander.

This is the site of a Māori pā (village) established by the Taranaki iwi Te Āti Awa, and the site where the first Europeans to the region landed their whaling boats in the late eighteenth century. In more recent decades, it has become the place of a national museum, library, and cultural archive named Puke Ariki (Te Kāhui Whaihanga/New Zealand Institute of Architects n.d.), which means "hill of chiefs" in Te Reo Māori.

*This is where we meet. Yes, Boulder, I am referring to you.*

In keeping with colonising settler habits, it is a hill that became the seat of the colonial government.

My evening visits to local pubs draw stories from town residents about the building of this grand scheme[1]. Some call it an exorbitant folly, a waste of taxpayers' hard-earned cash, and a vexing to the god of the sea. Others extoll the economic value of bringing jobs, notoriety, and tourist income to locals, a message already well-worn in support of the offshore oil drilling companies. One person recollects the interminable rumble of trucks through the city, and the percussive thunder of heavy loads of boulders being dumped, a percussion he says he still feels in his belly.

Just north of New Plymouth, one finds several quarries which were all contracted to harvest, transport and spread the boulders of the coastal walkway. An imaginary hum of diesel engines and a trail of muddy truck tire tracks lead to Jones, very likely your previous resting place (Jones Quarry n.d.) (Figure 1). There, along the edge of the Mangaoraka Stream, a stream snaking through the alluvial soil, to the sea, to the sea, your aunties, cousins, and uncles lodge in a tight matrix blanketed with layers of alluvial soil topped by native grasses. Buried together as deep banks, your ancestors are sleeping, like you did until the jaws of a digger cut a vivisection to this bedding, yanking drowsy toka, stones, dismantling embraces and disturbing dreams. A geological columbarium disinterred.

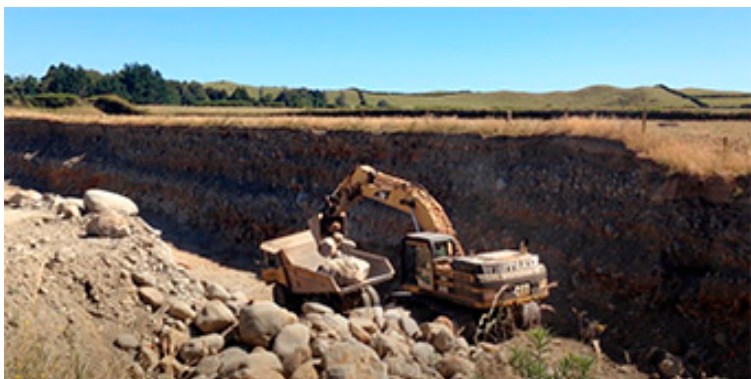

**Figure 1.** Boulders being harvested at Jason's Quarry. Still image from video courtesy of Stuart Foster 2015.

Grey as grey, the stones, the sea, the sky, and the rain are visible in stark contrast to my alien white flesh. The gloomy atmosphere of the late summer day dampens the celebratory nature of the 2015 Water*Peace Walking Symposium, a promenade from the native bush to the sea along which the public could engage numerous art works (Intercreate.org 2015). A few non-weather-averse people walking along the coastline hold their heads down and their shoulders up to ward off the wind; they follow their feet rather than the horizon.

### 3. Approach

Fear of losing my footing means approaching you is a slow methodical balancing process far from straight-lined. My feet lack the necessary suckers of frogs, slugs, or squid to dance nimbly down to the water; I am not that well-evolved. There were no prior rehearsals, strategies or scripts to initiate this encounter; it is complete liveness; everything happening in the moment, in continuous space, real time (Lindelof et al. 2016, p. 229).

Of all the boulders at the mouth of the stream, why you? I suppose it is the fullness of your form, the way you seem so stoically stuck into the sand and resistant to the tide, slightly defiant (Figure 2).

*No, you seem sure, confident, apart from the others, solo in the presence of others, rebellious or reticent. No, you are just a rock.*

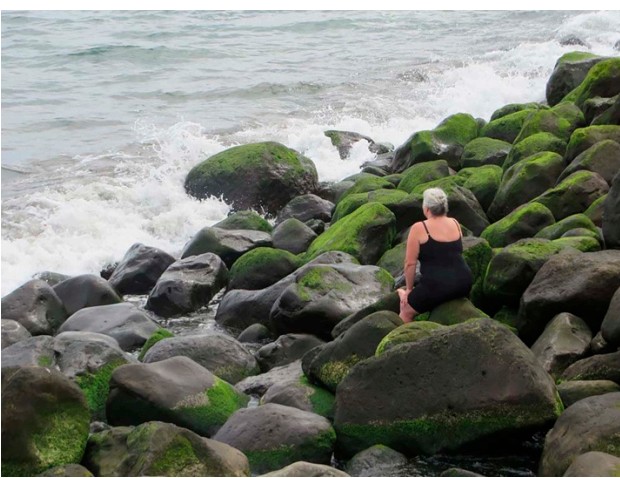

**Figure 2.** Initiating the approach. Image by Nina Czegldy 2015.

My own full form feels affinity to your shape and its huggable, embraceable proportions and its dark, molten, slightly scarred surface, a surface far more nuanced than my black togs stretched across pale pink and angry red sunburn flesh. It is unclear if your shine is rain, sweat or the polish of age. You are here, I am here, not touching, but within reach.

*Boulder, are the plumes of my warm exhales melting your demeanour?*

### 4. Apart in Close Proximity

A rogue wave crashes over our bodies, catches us by surprise, breaks the muteness. Your staunchness exceeds the weakness and fragility that is my own humanity. You are made wet again and again over many years, unlike me, shivering in the descending wind chill factor. On its diminished return, the sea carves out a hollow space at the base of your belly, sucking, digging, hoping that this time it will win. Feet sinking into its clutches, I notice my anthropocentric penchant to personify emerging. Ascribing human features and qualities may in fact be an insult. To appease the affront, I admit that I have a heart that oscillates from gold to stone cold, so perhaps there is some stoniness in my ancestry after all. My square stature anchors me and my wandering lofty thoughts to this world, the earth (Figure 1).

*I wonder what you might say about me if you could speak, speak as I do that is.*

Transient . . . sporadic . . . short lived . . . hurried . . . swift . . . small . . . breathless (Cohen 2015, pp. 30–33). Caution. "Slow down . . . Or, should I say, I am speaking *to* it? Or should I say, it is speaking *to* me? . . . [M]inding place poses the very problem of contact and how things relate . . . my very writing (right now!) is an alliance . . . that takes us to the weird joys, strange horizons, and new modes of being that complicated assemblages afford . . . [not] an act of prosopopoeia . . . [but] an ethics of interdependence" (Duckert 2012, p. 277).

### 5. Looking Closely

Resisting an ingrained impulse to touch you, I hold fast to merely stooping over, to look closer and in more detail—to follow the practice of naturalist John Muir to take the time to become acquainted with everything; to try to hear what it had to say, to ask a boulder "whence they came and whither they were going" and then, to follow (Duckert 2012, p. 278). Spoken to and heard from. Slow down to take the time to notice, to record, to follow "the complicated relationships between things—and, at all times, to address their grievances . . . *hello everything*" (Duckert 2012, p. 279). Looking down (at you) to look up (to you).

*What is the story about that gash running across your mottled surface? Do you feel pain?*

### 6. Emulating

In my head, I am crouching into a tight ball, the smallest volume this plump body can muster. Knee-hugging, back hunching, head burying, every fibre of my being is teaming to emulate you as a stubborn statue. I meditate on geological processes based on compression—all stress is directed to the centre, squeezing, folding internally, on the verge of fracturing, bending, or crumbling (The Geological Society n.d.). Exhausting all airspace, all surfaces in full contact of one another, moving in slow time, the rock of ages, poised in petrification. I am determined to be still for as long as possible, eyes shut, consumed by interiority, to assume an empathetic posture, in silence, to blend in, endure the pain, pass the time, which escapes measure other than the countless waves that test my resolve. My tailbone is pressing into the hard curve of your exterior; it provides leverage. A tingling sensation, spinal energy, plugged into your liveness. Far from abstract or conceptual, this is a sensuous mimesis; a human body co-mingling with its environment; a human body foiling the historical trap of figure-ground, escaping the 'world-picture' that makes humans the master of that world (Levin 2014, pp. 7–8). Practicing geo-mimicry (Figure 3).

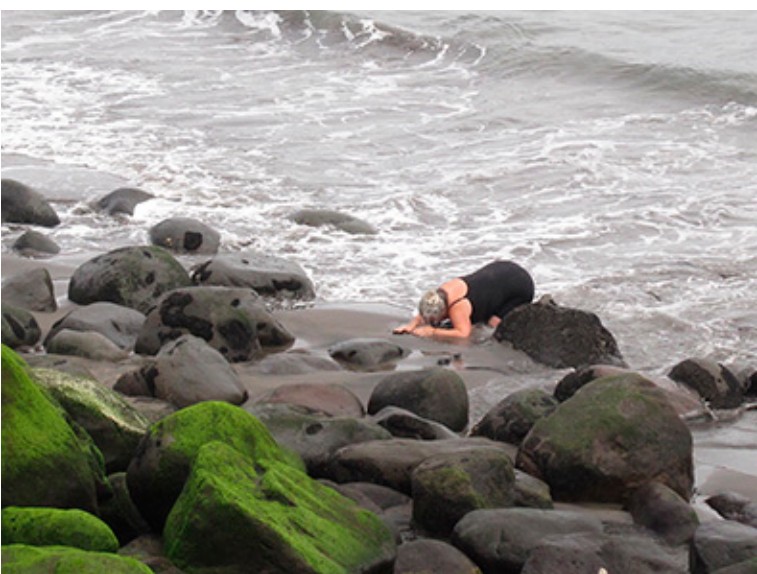

**Figure 3.** Mimicry in action. Image by Nina Czegldy, 2015.

For some, mimesis is less than honourable; to copy, represent, assimilate, even appropriate; to fly in the face of being original, authentic, and genuine. And yet for other creatures such as plants, plants, insects, fish, reptiles and animals, mimesis as a form of camouflage is a skill to survive, sustain, and mate. To mimic is a high-level form of empathy—to mentor, to teach, to train, to shadow, to follow, to "produce affinities an essential creative impulse . . . to behave like something else . . . to become ground" (Benjamin 1978, pp. 333–34). My awkward attempt to be a boulder is less an attempt to become invisible as it is to take refuge in the spatial collapse that is camouflage (Levin 2014, pp. 6–11).

*So, asking you rhetorically, how much bolder can I become? And, what are those coastal walkers witnessing? Am I camouflaged? Can they see me, pick me out from amongst all the boulders holding back the sea, just another (paler) boulder beside you? Is my guile enough? Or too much?*

I whisper to myself, "Don't flinch". Feel that fly making its way up my arm like it belongs there. Enjoy the tickle; just feel it, listen to the sensation.

*Considering all these micro-dramas, how do you maintain such patience and composure?*

## 7. Raising Alarm

The next wave bowls me over. The world spins. This human-stone unfurls to expose its soft underbelly interior; the ruse is up. And the top to my togs is down. Flashes of blinding whiteness. Voices are shouting and showering worry, concern, and alarm towards me, now a bundle of appendages flailing in the surf (Figure 4).

*No, police officer, I am not mad, nor demented, suicidal or self-harming; I am just performing, just becoming a boulder, just a boulder. It's art; it's just art.*

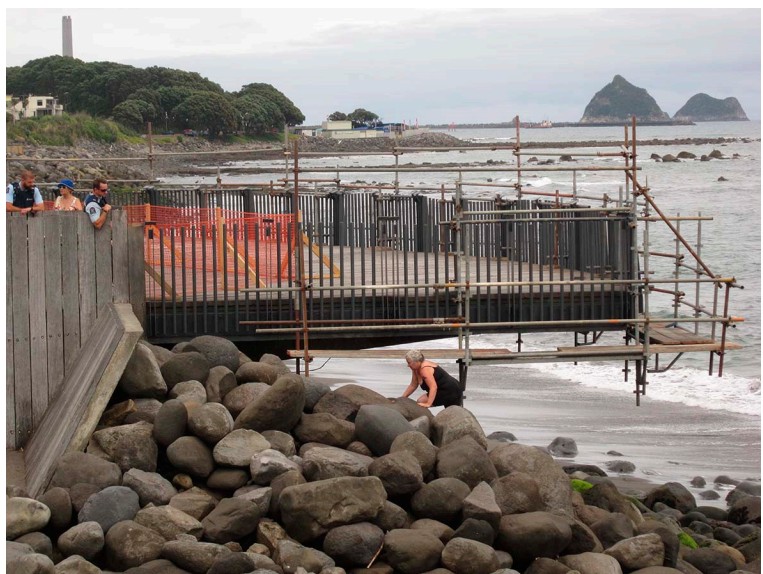

**Figure 4.** Addressing concern by the public. Image by Nina Zegley, 2015.

Their fears appear appeased. They are walking away unfettered by the unfathomable interaction you and I are having. Their misinterpretation of the scene will haunt me for the rest of the day, and the years to come. I worry that I caused some one distress, unintentionally or not, which niggles at my over-arching desire to practice ethically with humans and other materials. I wonder what it will take for an audience or spectators to discard age-old visual habits based on figure: ground relationships for fully immersive, non-hierarchical ecologies? "What might it mean . . . to present the self not as an atomised individual moving within the environment, but rather as the environment itself, something that is coextensive with its surroundings?" (Levin, p. 6). Our encounter is a hybrid form of eco-ethics (Patrizio 2020) whereby our interactions are guided by moral principles governing our human: environment attitudes and a mutual concern for each other's care and preservation (Brennan and Lo 2021).

## 8. Facing

Gathering composure, kneeling, fronting the sea to be forewarned of the next rogue swell, face to face. Forehead and nose are pressing against some fine furry algae creeping over your eastern-most face. Two hard mineral heads knocking, the calcium of bone and the collagen of cartilage sharing whispers with fine-grained apatite, garbet, ilmenite, biotite,

magnetite, zircon, and trace amounts of alkali feldspar extruded violently as a glassy matrix from the belly of the earth (Geology.com n.d.). So, your outer firmness belies your inner liquidity, or rather, as a sensual object, your frosty (Figure 5), "swirling superfluous outer shell" is a cover-up for the molten inner core "…where sentience takes place" (Harman 2007, p. 179). You smell of dead fish, bird guano, the stale sulphurous odour of bacteria that feast on phytoplankton, the sex pheromones produced by seaweed eggs, and the iodine stench of "bromophenois produced by marine worms and algae" (Villazon n.d.). The anxiety of the last few hours means that my robust musk-meets-garlic fragrance is liberally wafting your way.

*It might be love. At the very least it is some form of geophillia.*

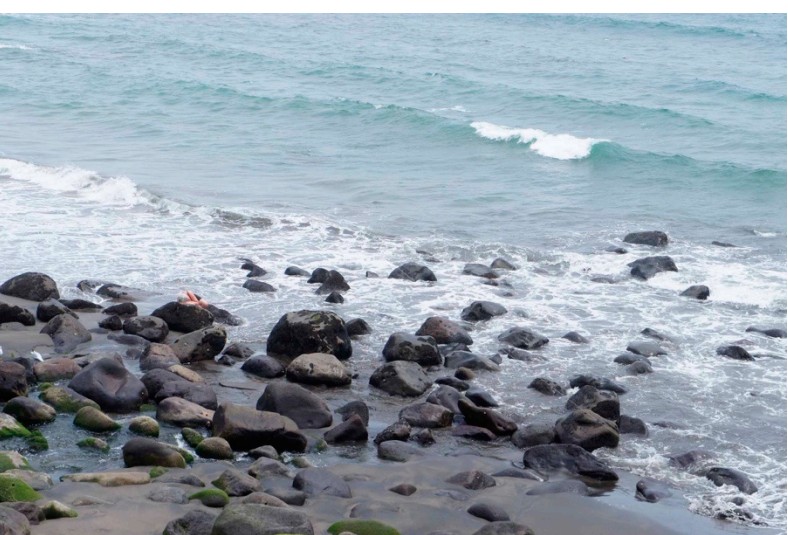

**Figure 5.** The pressing of flesh to boulder. Image by Nina Zegledy, 2015.

### 9. Venting

The sun is cutting through the haze. Its position and brilliance offer an inkling of how much human time has elapsed give or take an hour. The proscenium-like mist draped over the scene is evaporating. And while admiring the faint and feint gasps of thermal vents rising from you and all the other boulders around me, I am falling into a gentle swoon that is my own miasma. The taste of ocean air renders lips to dry and crack while it decorates your parched curvature with a pale white scaley powder. Perhaps there is value in the Biblical saying, "You are the salt of the earth" (The Bible (King James Version). *Matthew* 5: 13). Perhaps it applies to both of us equally? Worthy, virtuous, solid, dependable, honest, unpretentious mineral bodies emerging from a dead sea (Grammarist 2022). Sharing this cocktail of calcium, iron, and phosphorous is our mineral moment of intimacy (Ellsworth and Kruse 2013, p. 17). "Like stone, human flesh mingles dry earth with binding water; an unsettled union of wet and dry, cold and warm, fire and tears" (Cohen 2015, p. 1).

And yet you are so impenetrable, so profoundly frustrating, recalcitrant, ubiquitous, shrouded in lithic obduracy! (Cohen 2015, pp. 16, 50). My very presence seems to have no effect; you are not moved in the least. You simply endure in blunt reality (Cohen 2015, p. 16). Your greatest achievement is to simply exist, firmly, which, as resistance to time is your power and agency (Latour 1993). Living at two vastly different scales of time, two durational temporalities, this rapport reminds me of the way my partner and I lived for years, out of synch, out of pace, as salty stoic bodies in a mundane, dilatory romance (Cohen 2015, pp. 10, 29). Might this be geophilia—the unrequited love of stone, "an urge to affiliate with other forms of life, to natural love of life for life, matter's promiscuous desire to affiliate with other matter regardless of organic composition or resemblance to human vitality" (Cohen 2015, p. 27)?

*Not sure how to interpret that stalwart and seemingly aloof stance you maintain.*

## 10. Mouthing

*May I stretch out across your breadth to lose these goose bumps to the sun dogs?*

Consider me a fish stranded on the island that you pose, unflappable dehydrated flesh, a vain human body desiring warmth and a tan camouflaged as a sacrifice to the seagull species that are hovering, diving, and screeching at an alarming pitch. Splayed, my innards attract increasing aviary attention. I may be on top, but the dominance is yours as the solid rock plinth that serves up the monument, or in some cultures, the sacrifice. I am at your mercy as an organic geosexual statue, a perched representation of human-centric values. It is an uncomfortable posture. I am awkwardly visible, illuminated, out in the open, on an altar, exposed for all the hubris my kind has brought to this planet, which, at this moment, overshadows all the love it has generated (Figure 6).

*Let's remember that we are at the mouth of a stream where boulders such as yourself have been placed to keep the mouth from wandering.*

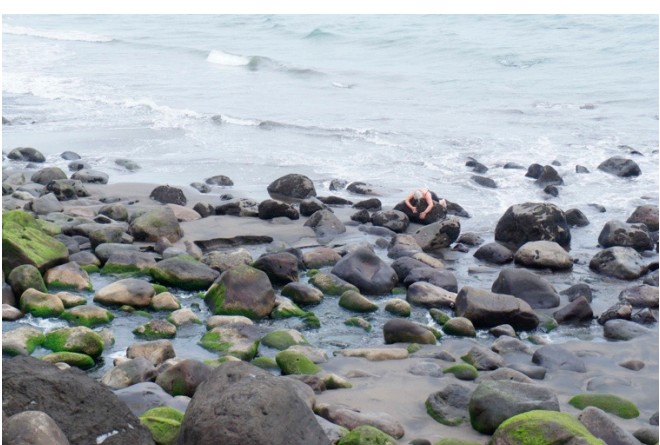

**Figure 6.** Sun fodder. Image by Allan Giddy 2015.

The thought of a wandering mouth begets lucid dreams of stony beings. Stone and statues are ancient bedfellows. Fond memories flood into focus of trapsing through Greece as a young architect. Ideal marble bodies surfaced in milky veils to hide (or not) nipples, phallus and pubic regions stare back with blank eyes blinded by past and future atrocities. As if filled with embalming fluids. Weighed down by their burden to symbolise or to add semiotic significance. To endure the ages. And yet, in the piercing heat of summer, their pursed lips part ever so slightly, and the inert mass becomes something different, animated, vital, a living, breathing, speaking, active, conscious, sensible, thinking subject (Serres 2015, p. 87) (Figure 7).

*Where are your lips? How do you speak? What do you speak? How do you feel, touch, hear and taste? Differently to me, most certainly. What secrets and secretions do our different mouths salivate and swallow?*

*I am searching for hints of your subjectivity so that I can write to you, not about you.*

To become with you (Haraway 2008, p. 244). To become boulder. Seeing colour return to your skin (Serres 2015, pp. 88–92). So, we are both stones and statues.

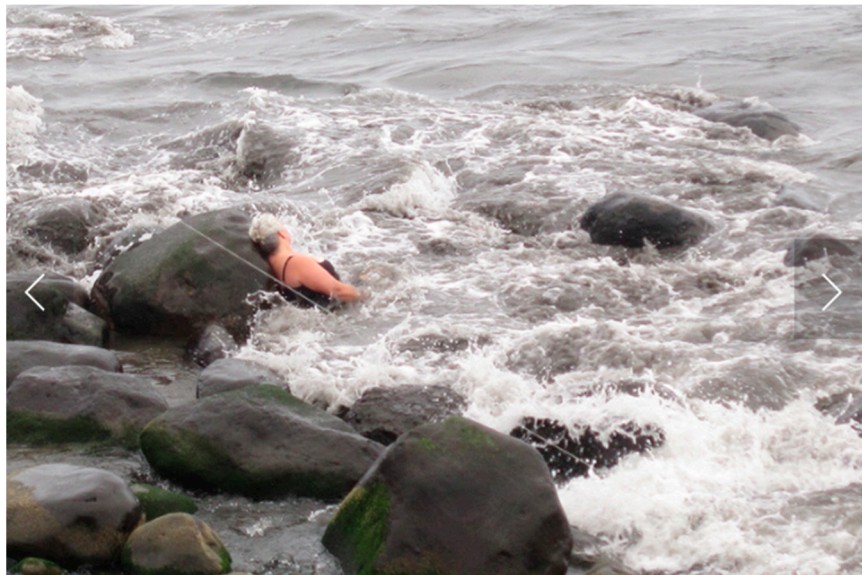

**Figure 7.** Swamped. Image by Alan Giddy, 2015.

## 11. Immersion

We forget to watch the horizon. Our radar is knocked off kilter as are our bodies. The wave leaves no escape as it throws itself at the shore. What lurks in the background now takes over the foreground. It is all ground, all immersed ground, literally. I am clutching, grasping, wrapping my legs and arms around your bulk. You are anchored by rocky sediments unlike my emotive sentiments which cast me adrift from survival instincts other than to hang on <u>for</u> dear life. You are life. Long life. Hanging on <u>to</u> dear life. The pragmatic reality of being mortal is all too apparent. It is fraught with fear of failing, falling, dying. While living, it is mine in which to be mired. It infers a thread between life and death, human and non-human, where life is worth fighting for, grasping on <u>to</u>. Funnily enough, at this moment, floating is not an option. Unlike you, I am not made to survive under water.

*Help. Please.*

I promise to defend your petric agency and integrity when your species is cast as dumb, mute, deaf, speechless, still, lacking, uncomprehending, stupid, insentient, impotent, or deceased (Cohen 2015, p. 50).

*Help.*

We are robbed of only a few breaths, less than a minute. Hardly enough to justify the sputtering gasps. The current is dragging my dignity into the swift current as large white breasts bobble in the surf without censor. Now we are both naked. Both alive; more and less petrified.

## 12. Heat

Adrenaline mixed with hypothermia and tears are unsettling me as I clamour up to the walkway not looking back at the scene of our liaison. It is the nature of durational work not to keep, watch or note time and so it follows that re-entry to the pedestrian world is an emotional shock. A sweet embrace of a dear friend who knows the risks I take and what is at stake by performing here, today, in this place, as an alien, and a foreigner to this land soothes my sobs. She knows my tenderness, foibles, shortcomings, and power, just as you do. I sob into her shoulder, a sign of the deep transformative agency of our relatively brief encounter (Yates 2002, p. 50) (Figure 8).

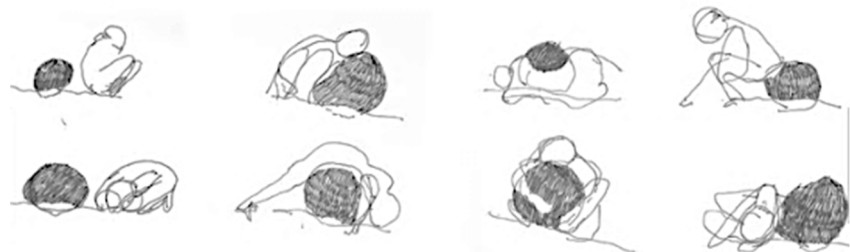

**Figure 8.** Drawing still moments of improvisation. Image by Julieanna Preston, 2015.

### 13. Musing

I rarely make a habit of parsing the world using definitions or taxonomies, which are often assumed to be truths with hard, impenetrable edges, as if they were immoveable, unassailable, non-transformable rocks. As this essay muses, boulders (and definitions) seem to mark out an ambiguous territory of associative notions and concepts that in their imprecision denote qualities rather than identities; they suggest porous subjective boundaries rather than discrete objects. Such is the agency of geo-poetry. This is most certainly the case for disciplines of even creative practice, or better, of praxis. It follows, and with all respect, what is generated by entertaining Art and Performance as discrete or conflated or having a shared interdisciplinarity? Does not that mode of compartmentalising knowledge just repeat anthropocentric ideals of hierarchy and ownership? Here I bear out my own staunchness as a disciplinary agnostic.

*You are not the least bit concerned for this divisive human folly, are you?*

My artistic research is more concerned with what one can do than what category or form it takes. The knowing is found in the doing and in the space of actions. The performance of "Becoming Boulder" has no script and there are no plans or scheduled to regulate chance, serendipity, or contingency. The performance is merely a spatial envelope of sensing gestures, usually mute/ nothing spoken, that, as an embodied experimental inquiry, unfolds in unexpected, unimaginable dimensions. Sound, taste, and hapticity tend to overwhelm sight's typical primacy. The performance occurs amid and with everyday life, not on a stage, but enmeshed with all the noise of quotidian life and often going unnoticed, with no specified audience alerted or invited.

This essay is meant to complement the performance, and yet, runs a different course. While it is a dialogue between me as the performer and the andesite boulder to iterate the original encounter as a site of inquiry, it is also written for another audience, you the reader. The essay performs a tri-part dialectic. Its written form effectively applies a structure to a script that organises the performance experience into phases of encounter narrated in a linear sequence if only by virtue of English-language writing top to bottom, left to right. To write spatially is a challenge to the conventions of journal articles with expectations of introductions, literature review and findings. To write an essay as a site of site-responsive critical reflection and expression invites, you, the reader, to put aside a hard and fast argument or supposition and to yield to the process of writing (and reading) as another form of wondering, wondering as it happens. Language lives.

In writing this essay, the method of situated learning continues.

*Has it been that for you, Reader?*

Such material situated learning is oblivious to binaries that hold humans at bay from the environment, especially figure: ground. In the essay, anthropocentric habits brutally show themselves: concerns for setting the scene, conjuring narrative imagery, managing metaphors, choosing words for their sonority, and admitting/ confessing the intrepid limits of human to boulder relations (no matter what level of mimicry was mastered). They register generative potential as much as obstacles towards casting aside long-standing constructs that place more value on some things and some people more than others and get in the way of acting on geophilia. Such separation from the material world highlights one

of the lessons that being here, in this place, amongst its history and people starts to erode, contest, or doubt. Whether it be live or in the voice of the written form, the improvisational gesture—the mere attempt to meet, touch, and become entangled—invokes a relational, co-extensive, co-existing collaboration with other material stuff; this is all that matters. Hanging on <u>to</u> dear life.

**Funding:** This research received no external funding.

**Acknowledgments:** Appreciation for their time, thoughts, comments, guidance, and inspiration extends to Kura Puke, Stuart Foster, Ian Clothier, and Inaaha Te Urutahi Waikerepuru. Thank you to Madaleine Trigg for introducing me to alternative forms of contact improvisation. Gratitude to the College of Creative Arts, Massey University for research funding that enabled me to take up this residency and realise this performance work.

**Conflicts of Interest:** The author declares no conflict of interest.

## Note

1　Recounted conversations with residents over three nights at Icons Sports Bar and Café, Peggy Gordon's Celtic Bar and Shining Peak Brewing, all located in central business area of New Plymouth, New Zealand.

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
