# Peer review of "Musing with Petric Bodies, Hanging on to Dear Life"

_arts, 2022_

Round 1

Reviewer 1 Report

Many congratulations, this is a beautiful body of work, both the performance and the essay. I propose minor corrections, only to allow this highly original work, to elevate its significance in the field of performance. My notes for minor changes below.  I hope my thoughts help define this incredible work, further:

Edit the abstract slightly, long sentences and repetition of ideas, could be set out a little clearer. The author calls the work an original essay, but as opposed to what? An unoriginal? – Can this be re-phrased to suit something on the lines of  a “critical reflection of practice”.   Be clear in this abstract, on what the essay is formally and conceptually doing and what the performance is doing/did.  For example, this long sentence: 

 “Framed as a series of contact improvisations between a large andesite boulder and myself, in a king tide, on a stormy day, at a culturally significant place in New Plymouth, Aotearoa, the essay employs emphasis and practices geo-poetry to colour the scene of musing and critically contextualise the potentials and limits of empathetic engagement with another kind of organic assemblage.” 

 I would suggest a slight re-structuring the above to make sense of what the essay is doing in relation to the performance, for example, it could read “ The essay employs ekphrasis and practices geo-poetry to colour the scene of contact improvisations between a large andesite boulder and myself, in a king tide, on a stormy day, at a culturally significant place in New Plymouth, Aotearoa. It will muse and critically contextualise the potentials and limits of empathetic engagements, from the performance with another kind of organic assemblage.”  

I would also suggest the words “another kind” is identified at this stage, name another kind, or if not, simply say “from the performance”.

Opening paragraph: Can you say where this mountain is in the world?

Line 33: “This is your petrogenesis, your creation story, or at least 33 one of them”. Who is the author addressing here? Is this creative text? If so, perhaps separate out to stand alone, so it is clear it is direct address. 

Line 89: “This is the atmosphere and context of a durational performance Becoming Boulder.” I would suggest this line would frame this detailed description of this beautiful landscape, articulated so well here, if it fore fronted the description, not as an after-thought. 

Line 96: I am excited by the direct address here… “Of all the boulders at the mouth of the stream, why you? I suppose it is the fullness of your form, the way you seem so stoically stuck into the sand and resistant to the tide, slightly defiant. No, you are sure, confident, apart from the others, solo in the presence of others, rebellious or reticent. No, you are just a rock…”.  But I would suggest the creative texts, that ‘speak’, are separated out after line breaks, are intended, or certainly have their own place peppered throughout – given space to breath. Italicised? This is a formatting issue that could allow the playful to be charged with the different kind of energy it deserves.  Whilst you have separated out the improvisations with titles,  which allows this textual practice its own space, the change to first person needs addressing – creatively – as another separation, so this voice reads as spoken/performative. 

Each improvisation is beautifully described, it is a detailed piece of work. I wonder however, if the format of each improvisation could be consistent? Some entries are longer than others, a mixture of prose and poetic text. I would suggest some spacing, to let the words breath and for the language to ‘trickle’. If the publishers permit this of course. If not, then perhaps edit slightly each improvisation for consistency of voice, as in some improvisations the first person voice disappears.

The section Post-Performance misses an opportunity to reflect critically on the relationship between the performance and the essay – and what the essay is doing. It touches the surface, but it makes a contradiction, here: “My artistic research is more concerned with what one can do than what category or form it takes. There is no script and there are no plans drawn or scheduled to regulate chance, serendipity or contingency when I initiate a durational site-responsive performance as an encounter with other material bodies. The performance is merely a spatial envelope of sensing gestures, usually mute/ nothing spoken, that, as an embodied experimental inquiry, unfolds in un expected, unimaginable dimensions.”  I notice here that the contradiction, lies between what the work is (performance) and what the essay is literally doing – (making form). Yet the form itself, can’t be denied, rather it should be acknowledged that the 9 improvisations of reflections is a script  - it is a voice and it is doing the job of imaginatively contextualising, reflectively, it is, performative – which is exciting. But the writing needs to acknowledge this form and its direct relationship to the live experience to make each component distinct and interrelated. The images are very affective, moving, embodied, and the reader, I imagine,  would want to know a little more about how these experiences, specifically, writing about them here, creatively, reflectively, contribute to a dialogue that you propose early on in the essay.

Author Response

First, it is rare that I get offered such compliments and advice from a peer reviewer, so I am exceptionally grateful. Your comments along with those of the other reviewers are some of the most help I have every received. Your review reveals to me that you are in tune with the essay and performance.

I am responding to your points in the order they were presented in the review report.

  1. I have edited the abstract, first, to shorten the sentences and remove some of the redundant text. Though I was initially resistant, I inserted factual citation data about the performance and adopted your advice re "from the performance".
  2. The opening paragraph now clearly states the location.
  3. What was originally line 33 has been cleaned up to make evident who 'you' pertains to. In fact, the next comment assisted to help resolve that problem you saw.
  4. What was originally line 96 regarding the voice of direct address. has been changed throughout the essay as a stand alone paragraph and italicised. I will see what the publishers/editor have to say about this as it stands outside of the journal's style guide standards. Additional bits of this italicised direct voice have been added judiciously throughout the rest of the essay to help maintain that distinctive energy you mentioned.
  5. It was deliberate to make each of the improvisations different in length. While the original version numbers each section as an improvisation, I have done away with the numbering in an effort to have the entire performance discussed as one long improvisation with various phased encounters which helps to de-objectify the gesture of each section. I have taken up your encouragement to 'trickle'. We will see how that flies. As mentioned above, the first person voice now occurs in each section.
  6. In the original manuscript that you reviewed, the Post-Performance, I felt was a bit weak. I have amplified the emphasis on situated material learning, highlighted the issue's focus on Art & Performance, revisited the cultural context of the performance, and as you noted so wisely, elaborated on the relation that the performance and text have to one another.

Thank you.

Reviewer 2 Report

“Musing with Petric Bodies, Hanging on to Dear Life” offers a site-specific context for the durational performance Becoming Boulder, focusing primarily on an analysis of the process of ten improvisations. The performance occurs in New Plymouth, Aotearoa, which the author stories through Indigenous, settler heritage, and scientific knowledges. Through the analysis of process, the author identifies the moment in the second improvisation when anthropocentric descriptions surface in the human-rock relations. In the fourth improvisation, analysis of process outlines the utility of (geo)mimicry while in the sixth improvisation— eco-ethics. Improvisations seven and eight explore a geophilia that gestures towards geosexuality. The article’s concluding contribution to knowledge asserts that a site-respondent praxis of becoming through performative improvisation foregrounds “sound, taste and hapticity” over the visual. The article as a written form performs the geopoetics of Becoming Boulder through word choices focused on sonority and similitude.

Nice combination of scientific references and Indigenous traditional knowledge. Within the introduction, I would find it of value to have the author (who will be the performer) introduced with any identity markers that situate them in relation to the land that is in focus. Do they dwell near the land? Are they a visitor? And what is their relation with settler or Indigenous heritage? Drawing out these elements of relationality will help to contextualise the narrating body in relation to the research shared. I do note that this is briefly addressed in line 85; it could be helpful to have this information earlier in the introduction, and expanded, so the reader develops a clearer idea of the article’s narrator.

I would love to have more information regarding the performance Becoming Boulder prior to the introduction of each improvisation. Over how many days/months/years has the project developed? Why durational? Who performs, for how long, etc.? General description of performance concept prior to analysis of improvisation process will help ground the reader in the larger context of Becoming Boulder. It could also be of value to situate the title of the work in terms of its ‘becoming’ in relation to Donna Haraway or other artists who engage ‘becoming with’ as research methodology.

It would be great to know an approximate length of each improvisation, what the artist wore, temperature/weather/date. It would be equally excellent to know what initiated and ended each improvisation, and how the improvisations were delineated as separate.

In Improvisation Four, would it be of value to introduce the terms biomimicry or geomimicry? What the author describes through discussion of mimicry recalls both, and either term would be well suited as a possible keyword for this article.

The conclusion includes: “With an aim to overturn many binaries that hold humans at bay from the environment, especially figure…” I would love to know more about this, as it sounds like the conceptual framework at the performance’s heart. It would be excellent to have this in the introduction, with expansion of what binaries the author has in mind, and also why there is an aim (or necessity) to overturn them.

In addition to the substantive suggestions I offer above, here is a list of copyedit suggestions for the author to consider:

Line 41: Should this sentence start with ‘Through (the natural processes of…)’?

Line 64: Possible to have a date here for when the whaling boats first landed?

Lines 69-74: Would be wonderful to have a source or sources that support these anecdotes. Either this, or indication that the author of the paper has first-hand experience with these conjectures.

Line 99: it’s should be its

Line 155: Is there a word missing in the first question? “What (are) those coastal…?”

Lines 157-9: Who is the ‘your’ referring to? So far, ‘you’ has been the pronoun assigned to the boulder with ‘I’ being the pronoun assigned to the author. In these lines that refer to human body parts, it is unclear whether the ‘your’ is of the boulder or the author. Consider, for consistency, shifting to ‘my’ if the author.

Line 161: bowls (not bolls)?

Line 162: underbelly (compound word)

Line 165: It’s (not Its art)

Line 278: The author mentions site-responsive here for the first time. It would be wonderful to have this included in the introduction, and potentially with a brief definition / citation.

Line 282-4: The information about audience and location of the performance would be useful to have outlined in the introduction as well.

Figure 2: There are left/right arrows on the image. Unless they are integral to the artwork, is it possible to source the photo without the arrows? The arrows’ presence leaves me wondering why they are included.

Author Response

Thank you very much for your comments, suggestions and critique.

I offer the following response in the order of your points:

  1. Geophilia/ geosexuality: This offering was a gift and I have woven these terms into the essay as they support the ekphrasis aspect significantly.
  2. Who is the performer and the performer's relation to indigenous people and place: I am well aware that this element was not in the version you read; I left it out because I felt that it would identify me to such a degree that it would reveal my identity and tamper with the required anonymity for the review. This has been inserted in the front section as a form of introduction.
  3. I have woven more of the details of the performance throughout the essay as suggested including a reference to 'becoming with' ala Haraway.
  4. I have resisted offering the time of each improvisation, and actually removed each improvisation section being numbered. Here I am trying to soften the boundaries between different events in the performance as if to break it down into discrete movements, with a score or timetable. This is mode of counting and measuring is contrary to my practice.
  5. I have incorporated the term geo-mimicry as suggested.
  6. I have reconsidered the overturning of binaries that you interpreted as at the core of the performance. this is true for much of my work, but having read the manuscript gain with time to reflect, I feel it is too big and heavy-handed to tackle in this essay; I understand it as a long term goal and a value but not the heart of the work.
  7. What was line 41 has been corrected.
  8. A date has been added to what was line 64.
  9. Lines 69-74 has been attributed to the conversations that generated those anecdotes.
  10. What was line 99 has been corrected.
  11. What was line 155 has been corrected.
  12. I have clarified the pronoun identifier in what was line 157-9.
  13. What was line 161 has been corrected.
  14. What was line 162 has been corrected.
  15. What was line 165 has been corrected.
  16. I have added a few sentences in the first section (what was called Introduction) to elaborate on site-responsiveness.
  17. I have added information about the audience and location to what was formerly called Introduction.
  18. Figure 2 and arrows: While the manuscript has been under review I have located the original high resolution image to use. I have also been making watercolours to replace some images which I feel do not expand the essay's message or content.

Reviewer 3 Report

The article is well written on the whole. It presents interesting and compelling reflections, questions and a sense of poetics at times, that also operates effectively through the researcher's personal voice. 
There are several aspects that need addressing in the article:

- Considering the author is engaging with mātauranga Māori (pūrākau), mentioning the local pa and one of the local iwi, they/she/he needs to present their own position statement at the start of the article in the introduction somewhere (say around the middle of it for instance) - this should include their cultural identity, and their relationship to this geographic context and to talking through mātauranga Māori knowledge here (for instance, even as a visitor and to acknowledge and respect for local Māori iwi and their knowledge). This is fundamental and ethical. Plus this needs to be reflected on later on in the article, perhaps in the closing remarks/conclusion. It's also been implied int he abstract that this will be done too (but it's not in the article to any sufficient level). 

- The naming of Māori gods needs amending so that they are all accurate (eg. line 37 Pūtoto needs to be replaced with Rūamoko, line 52 Tū-te-wehiwehi needs to be amended because it is the god of storms, lightening and reptiles, not war. Best to double check this with both online reputable sources and someone from local iwi ideally - or if the latter is not possible, this can be done with a Māori colleague familiar with this kaupapa, as it is pretty common knowledge across kaupapa Māori circles).

- Māori grammar needs fixing on line 37 for Māori world view - te ao Māori.

- Need to spell pā with a macron and give a brief definition of of it (line 63). 

- The abstract uses the term 'contact improvisation' but the article does not - I suggest that either the author unpacks this term in a sentence in the article (including in relation to the contemporary Western dance genre of contract improvisation) or does not use it at all and amends the use of the term in the abstract. 

- There needs to be an addition of a clear description of what the artist/author did in each 'improvisation' section - this could be done with a covering few sentences in the introduction that describes in first person what she/he/they did, what was performed in each section and what their artistic research approach/critical practice/methodology was within these 2 sentences. There also needs to be a clarification of what they/he/she mean by improvisation in a sentence and to qualify it with a reference. 

- Need to double check the referencing - just in case the referencing is not what the publication requires (eg. '(Geonet)' (It may be fine I am not sure, and if it is please ignore this). 

Once all of the above is addressed it will be good to publish I suggest. 

Author Response

Thank you for your review. Your comments have been very helpful. I respond below to your points in the order they were delivered:

  1. I made a decision to not include a statement about my relation to the place, people and land with respect to mātauranga Māori because I felt that it would violate the anonymity required for blind peer review. Yes, it is fundamental and ethical. I understand your comment; this matter has been inserted and attended to significantly in what was previously named as the introduction. I have followed your advice to come back to this point at the closing remarks.
  2. I have corrected  what was line 52 to name the correct god of war. However, in my original research and consultation with several Māori colleagues, the god of magma was confirmed as Pūtoto. 
  3. What was line 37 has been corrected to Te Ao Māori.
  4. What was line 63 has been corrected as pā and described accordingly.
  5. What was called Introduction, now contains a paragraph specific to contact improvisation including a reference.
  6. I have reconsidered structuring the manuscript and numbering those sections as discrete improvisations. This corrects the sense that there were ten improvisations when in fact the performance was one improvisation that evolved over the duration of it. Each portion of the manuscript signals what the gesture or movement was and its implications.
  7. The referencing format has been reviewed and now fully adheres to the journal style guide.